# Recent Developments in the Interactions of Classic Intercalated Ruthenium Compounds: [Ru(bpy)_2_dppz]^2+^ and [Ru(phen)_2_dppz]^2+^ with a DNA Molecule

**DOI:** 10.3390/molecules24040769

**Published:** 2019-02-20

**Authors:** Fuchao Jia, Shuo Wang, Yan Man, Parveen Kumar, Bo Liu

**Affiliations:** 1Laboratory of Functional Molecules and Materials, School of Physics and Optoelectronic Engineering, Shandong University of Technology, Zibo 255000, China; wangshuo9414@163.com (S.W.); kumar@sdut.edu.cn (P.K.); 2Beijing Research Center for Agricultural Standards and Testing, Beijing Academy of Agriculture and Forestry Sciences, Beijing 100097, China; many@brcast.org.cn

**Keywords:** intercalated ruthenium compounds, molecular light switch, sensitive luminescent reporter, binding mode, single molecule force spectroscopy

## Abstract

[Ru(bpy)_2_dppz]^2+^ and [Ru(phen)_2_dppz]^2+^ as the light switches of the deoxyribose nucleic acid (DNA) molecule have attracted much attention and have become a powerful tool for exploring the structure of the DNA helix. Their interactions have been intensively studied because of the excellent photophysical and photochemical properties of ruthenium compounds. In this perspective, this review describes the recent developments in the interactions of these two classic intercalated compounds with a DNA helix. The mechanism of the molecular light switch effect and the selectivity of these two compounds to different forms of a DNA helix has been discussed. In addition, the specific binding modes between them have been discussed in detail, for a better understanding the mechanism of the light switch and the luminescence difference. Finally, recent studies of single molecule force spectroscopy have also been included so as to precisely interpret the kinetics, equilibrium constants, and the energy landscape during the process of the dynamic assembly of ligands into a single DNA helix.

## 1. Introduction

Deoxyribose nucleic acid (DNA) is a very long, thread-like macromolecule built from a large number of dexoyribonucleotides, composed of nitrogenous bases, sugars, and phosphate groups [1]. The bases of DNA molecules carry genetic information, whereas their sugars and phosphate groups perform the structural role. All living cells on earth, without any known exception, store their hereditary information in the universal language of DNA sequences. These monomers string together in a long linear sequence that encodes the genetic information. Since after the introduction of the DNA double helix (B-DNA) by Watson et al., non-classic DNA structures, such as right-handed double helix with a shorter and more compact helical structure (A-DNA), left-handed double helical structure with a zigzag pattern (Z-DNA), DNA hairpins, triplex, DNA bulges, G-quadruplex, and C-quadruplex (I-motif), have sprung up (Figure 1) [2,3,4,5]. The diversity of the DNA structure and its different biological significance have encouraged scientists to use small molecules in order to explore different structures of the DNA strand, which is of great significance for molecular recognition, disease diagnosis, drug design, gene therapy, and so on [6,7,8].

In the past 30 years, ruthenium polypyridyl complexes have been widely tested in DNA binding studies, and have become the ideal candidates for the design of DNA binding systems [9,10,11]. There is a wide range of applications of ruthenium polypyridyl complexes in DNA molecular light switches, DNA structure probes, DNA-mediated charge transfer, and anticancer drugs, because of their richful spectral properties [12,13,14,15,16,17,18]. [Ru(bpy)_2_dppz]^2+^ and [Ru(phen)_2_dppz]^2+^ (Figure 2) as the classic intercalated ruthenium complexes are the most commonly used in probing the DNA structure. Since Barton et al. first reported the observation of a molecular light switch, the interactions between these two ruthenium complexes with a DNA helix have been intensively explored by the researchers. Several studies have focused on the mechanism of the molecular light switch [12,19,20,21,22,23,24]. The studies of the interactions between [Ru(bpy)_2_dppz]^2+^/[Ru(phen)_2_dppz]^2+^ and the different forms of DNA strands have been intensively studied by scientists for a better understanding of the photoluminescence of these two intercalated ruthenium compounds upon binding with DNA helix. The results showed a large difference in photoluminescence on binding with different DNA forms [16]. Hence, it is critical to determine the binding modes in order to understand accurately the large difference of photoluminescence induced by the different DNA forms. Although the determination of the binding modes and binding sites are critical for interpreting the luminescence difference, the binding kinetics and magnitude of the double helix structural deformations during the dynamic assembly of DNA-ligand complexes are still unknown. Herein, to get a deeper insight of the classic intercalated ruthenium-DNA interactions, the studies probed by the single-molecule force spectroscopy are also introduced. Therefore, in this brief review, we will mainly concentrate on the following four parts: the (1) mechanism of the molecular light switch, (2) sensitive luminescent reporter of nucleic acid structures, (3) binding modes, (4) and single molecule force spectroscopy study (SMFS).

## 2. Mechanism of the Molecular Light Switch

In 1990, Barton et al. first observed that there was no photoluminescence for [Ru(bpy)_2_dppz]^2+^ in an aqueous solution at an ambient temperature, but it displayed intense photoluminescence in the presence of double-helical DNA, which was known as the DNA molecular light switch [12]. [Ru(phen)_2_dppz]^2+^ also possessed similar photophysical properties and served as a “molecular light switch” for DNA (both complexes were optically isomerized as left (Λ−) and right (Δ−) handed enantiomers). Since then, their interactions with DNA have been intensively studied [16,24,25,26,27,28,29,30,31,32,33,34]. Several studies have focused on the mechanism of the molecular light switch, but no certain conclusion has been made till now [18,31,35,36,37,38,39]. The studies in the early decades revealed that both of the ruthenium complexes displayed no photoluminescence in the aqueous solution, but showed high luminesce in the presence of DNA. The luminescent enhancement observed upon binding the DNA helix was attributed to the sensitivity of the excited state, which could be quenched by water. Whereas, in the case of DNA, the metal complex upon intercalation into the DNA helix was protected from the aqueous solvent, thereby preserving the luminescence [12,13]. However, Turro et al. studied the binding of the dinuclear ruthenium complex with double strand DNA (dsDNA), and reported that the intercalation of the complex into the DNA strand was not the key to the molecular light switch; the dinuclear ruthenium complex bound with the DNA groove can also lead to the photoluminescence [40]. Meyer and Papanikolas reported temperature-dependent excited-state lifetime measurements of [Ru(bpy)_2_dppz]^2+^ in protic and aprotic solvents, and showed that the dark state was always the lowest in energy, even in an aprotic solvent, and the light-switch behavior was the result of a competition between the energetic factors and entropic factors that favor the dark state and the bright (bpy) state, respectively [30,31]. So, the currently highly accepted mechanism proposed by Olson et al., describes that the light switch effect of the intercalated ruthenium complexes benefited from the existence of two different metal-to-ligand charge transfer (MLCT) states—MLCT-1 and MLCT-2 (Figure 3) [14]. In the aprotic solvents, the MLCT-1 state was responsible for the emission, whereas in the protic solvents, hydrogen bonding lowered the energy of the MLCT-2 state, which became accessible and had a notably low luminescence quantum yield as a result of a rapid non-radiative decay pathway. To enlighten the global understanding of the molecular light switch, the binding of the ruthenium compounds to the DNA strands with different sequences and forms are depicted in detail in the following section.

## 3. Sensitive Luminescent Reporter of Nucleic Acid Structures

Ruthenium complexes are ideally suited for application as sensitive noncovalent probes for a polymer structure. The two complexes discussed in this review are often utilized to probe the structure of different DNA forms. Barton et al. exhaustively studied the effects of [Ru(bpy)_2_(dppz)]^2+^ and [Ru(phen)_2_(dppz)]^2+^ with different nucleic acid sequences and conformations [12,13,15,16,41,42]. They discovered the strongest luminescent enhancement for intercalation into different DNA conformations (B-form, Z-form, and triplex-DNA), which afforded the greatest amount of overlap with access from the major groove, whereas the greatest amount of overlap with access for A-form was from the minor groove. The triple-helical DNA showed the highest luminescence, followed by Z- and B-form helices, whereas the A-form exhibited the lowest luminescence. Differences were observed in the luminescent parameters between both of the complexes, which also correlate with the level of water protection. In the presence of nucleic acids, both of the complexes exhibited biexponential decays in emission (the emission characteristics of both of the complexes upon binding to the nucleic acids of the varying conformations and sequences are summarized in Table 1), indicating the presence of two distinguishable DNA binding modes for both of the complexes. The two intercalative binding modes for the dppz ligand from the major groove were proved via the following quenching studies: one where the metal–phenazine axis lies almost perpendicular to the DNA dyad axis, and another where the metal–phenazine axis lies along the DNA dyad axis [13]. Barton et al., also found that the luminescence of [Ru(bpy)_2_dppz]^2+^ was sensitive to DNA defects and RNA duplex defects [16,42]. In the presence of a single base mismatch of dsDNA (27 base pairs), large luminescence enhancements were observed for the Δ-[Ru(bpy)_2_dppz]^2+^, whereas Λ-[Ru(bpy)_2_dppz]^2+^ showed a particularly high luminescence when bound to an abasic site, which indicated the good selectivity of the isomers to the mismatched dsDNA. Both of the complexes can be seen as unique reporters of nucleic acid structures, and may become valuable for designing new diagnostics for DNA. The differences in the binding capacity of other DNA binders are summarized in Table 2.

Shi et al. systematically studied the interaction of these two classic intercalated ruthenium complexes with G-quadruplex (5′-AGGGTTAGGGTTAGGGTTAGGG-3′) and I-motif DNA (5′-CCCTAACCCTAACCCTAACCCT-3′), and discovered that [Ru(phen)_2_(dppz)]^2+^ had obvious light switch effects on the G-quadruplex and I-motif DNA, while [Ru(bpy)_2_(dppz)]^2+^ had light switch effects only on the G-quadruplex. By combing the results of the steady-state luminescence, circular dichroism, and UV melting, they found that both of the ruthenium complexes preferentially bound to G-quadruplex over the I-motif because of the difference in the binding modes. This was directly proven by the fluorescence polarization experiments, where both of the the metal complexes were bound with the G-quadruplex via the terminal π–π stacking, whereas for the I-motif DNA, only the non-specific binding was formed by electrostatic and hydrophobic interactions. Besides that, a luminescence difference also existed between these two complexes when interacted with the G-quadruplex and I-motif DNA. The reason for this may be that the ligand phen of [Ru(phen)_2_(dppz)]^2+^ contained a larger planar aromatic ring than the ligand bpy of Ru(bpy)_2_(dppz)^2+^, which was more easily combined with the G-quadruplex and I-motif DNA [17,44]. The results indicated that [Ru(phen)_2_(dppz)]^2+^ and [Ru(bpy)_2_(dppz)]^2+^ could serve as a prominent molecular light switch for the G-quadruplexes and I-motif DNA. This discovery of the binding features of G-quadruplexes and I-motif provides a more comprehensive understanding of their molecular recognition, which is valuable for designing new diagnostic agents and imaging agents.

In addition, McGarvey et al. also studied the light effect of these two kinds of intercalated complexes with single stranded DNA (ssDNA), and found that the luminescence intensity was significantly affected by the length of the DNA chain, and at least six bases of the DNA sequence can produce the light switch effect. They speculated that ssDNA surrounded the complex and formed a hole-like structure, which protected the complex from the water [36].

From the above studies, it can be summarized that [Ru(bpy)_2_(dppz)]^2+^ and [Ru(phen)_2_(dppz)]^2+^ have an excellent selectivity to different DNA structures according to the difference of luminescence intensity. The pronounced discrepancy in the luminescence quantum yield induced by the different DNA forms is closely related to the binding modes between them, which is very useful for designing a new probe of DNA structure.

## 4. Binding Mode

Intensive studies on the binding modes of DNA with [Ru(bpy)_2_(dppz)]^2+^ and [Ru(phen)_2_(dppz)]^2+^ via major groove or minor groove have been performed using a variety of spectroscopic techniques [13,45,46,47,48,49]. Until recently, several crystal structures have been reported, which provided direct structural evidence of DNA binding with these two ruthenium complexes, and revealed that all of these DNA intercalators are bound to the DNA through the minor groove [27,50,51,52,53,54].

Barton et al. reported the crystal structure of Δ-[Ru(bpy)_2_dppz]^2+^ at a high-resolution of 0.92 Å, bound to both the mismatched and well-matched sites in the oligonucleotide 5′-(dCGGAAATTACCG)2-3′ (underline denotes AA mismatches). Two crystallographically independent views revealed that the complex bound mismatches through metallo-insertion by ejecting both of the mispaired adenosines. Additional ruthenium complexes intercalated at well-matched sites, creating an array of complexes in the minor groove, stabilized by stacking the interactions between the bpy ligands and extruded adenosines (Figure 4). These structural studies attested to the generality of metal-oinsertion and metallo-intercalation as DNA binding modes [26].

Cardin et al. systematically characterized the crystal structure of Λ-[Ru(phen)_2_dppz]^2+^ and Δ-[Ru(phen)_2_dppz]^2+^ with different sequences of a DNA strand. A brief introduction of the binding mode between [Ru(phen)_2_dppz]^2+^ and the DNA helix was stated as follow [27,52,54,55]: (1) Firstly, in the crystal structures of Λ-[Ru(phen)_2_dppz]^2+^ with two oligonucleotide duplexes, the dppz ligand intercalated symmetrically and perpendicularly from the minor groove at the central TA/TA step of d(CCGGTACCGG)_2_, but not at the central AT/AT step of d(CCGGATCCGG)_2_ (Figure 5a,b). In both of the structures, however, a second ruthenium complex linked the duplexes through the combination of a shallow angled intercalation of dppz into the C1C2/G9G10 step at the end of the duplex, and the semi-intercalation of phen into the G3G4 step of an adjacent duplex. In each case, the complex intercalates DNA through the minor groove of B-DNA [27]. (2) Secondly, the crystal structure showed that both of the enantiomers of Ru(phen)_2_dppz^2+^ bound to a single d(ATGCAT)_2_ duplex and Δ-[Ru(phen)_2_dppz]^2+^ were better fitted to the right-handed DNA duplex, because the Λ-Ru(phen)_2_dppz^2+^ would clash with the nucleic acid backbone (Figure 5c). Both of the enantiomers intercalated from the minor groove in such a way that a phen ligand was stacked against a nucleoside sugar. However, the intercalation of the two enantiomers was distinct with different orientations. The orientation of the dppz was of great importance in understanding the different luminescence behavior of the two enantiomers; water has access to only one phenazine N atom in the Δ-[Ru(phen)_2_dppz]^2+^-bound DNA, whereas for the Λ-[Ru(phen)_2_dppz]^2+^, water can readily access both of the phenazine N atoms [52]. (3) Thirdly, in 2016, the first X-ray crystal structure of a Δ-[Ru(phen)_2_dppz]^2+^ bound to the well-matched DNA was explained. They showed how the binding site could be related to a more general pattern of motifs in the crystallographic literature, and proposed that Δ-[Ru(phen)_2_dppz]^2+^ can bind with five different binding modes (Figure 6), offering a new hypothesis for the interpretation of the solution data [54].

The binding of the homochiral Ru(phen)_2_dppz^2+^ to poly(dT*dA-dT) triplex has been investigated by linear and circular dichroism and thermal denaturation. The results show that the dppz wing of the ruthenium complexes intercalated between the nucleobases, thereby stabilizing the third strand. Because of the wing size effect on the melting profile, the other two phen ligands of the metal complexes were proposed to be located in the minor groove of the triplex poly(dT*dA-dT) [56]. Tan et al. performed the investigation of Δ- and Λ-[Ru(bpy)_2_dppz]^2+^ with triplex RNA poly(U)·poly(A)*poly(U), and found that Δ enantiomer intercalated into the triplex RNA, which displayed a significant ability in stabilizing the triplex RNA [57].

The determination of the binding modes and binding sites is helpful for understanding the mechanism of light switch and the large luminescence difference, but the magnitude of the double helix structural deformations during the dynamic assembly of the DNA-ligand complexes cannot be determined. However, this can be resolved with the development of single molecule force spectroscopy (SMFS), which can also reveal new details about the molecular mechanisms governing DNA intercalation.

## 5. Single Molecule Force Spectroscopy Study

SMFS study can provide deep insight into binding kinetics and the magnitude of the double helix structural deformations during the assembling of DNA-ligand complexes at a single molecule level, which is the major advantage over other techniques [29,58,59,60,61,62,63,64,65,66]. Before the use of this technique, the self-assembly process of DNA-intercalator complex molecules had been investigated via different experimental means such as thermal denaturation and stopped-flow techniques, which were employed to explore the first kinetics studies of DNA intercalation [67,68]. There were other early bulk approaches to study DNA interactions, such as nuclear magnetic resonance (NMR), gel electrophoresis, fluorescence spectroscopy, and linear and circular dichroism spectroscopy [69,70,71,72,73]. However these bulk studies of DNA intercalation were constrained by several experimental and systematic challenges, such as a limited range of detected concentration, and uncontrolled non-intercalative molecular processes [74,75]. The greatest advantage of the SMFS technique is that it allows for directly measuring the progressing dynamic assembly of the ligands into single B-DNA as well as non B-DNA structures, such as the G quadruplex, in real-time, with precise control of the experimental conditions [76,77,78]. Information about kinetics, equilibrium constants, the energy landscape, and the binding site can be extracted from SMFS experiments. Besides that, the SMFS technique has a wide range of applications in the protein–DNA interactions, measurement of covalent bonding, and protein folding/unfolding [79,80,81]. Nuñez et al. measured the intercalation of [Ru(phen)_2_dppz]^2+^ into the DNA strand with optical tweezers at a single low applied force. The thermodynamic parameters for the [Ru(phen)_2_dppz]^2+^ intercalation into DNA, and the occupation of the DNA helix were perfectly determined using the force data, with the help of the McGhee–von Hippel model. The affinity constant (*K*_b_) and binding site size were found to be (3.2 ± 0.1) *10^6^ M^−1^ and 3 ± 0.5, respectively [64]. Williams et al. investigated the DNA intercalation induced by Ru(phen)_2_dppz^2+^. The binding constant and site size were determined by measuring the ligand-induced DNA elongation at different ligand concentrations, using the optical tweezers technique under a different given force (Figure 7), and both were found to be strongly dependent on the applied force. The results showed that the applied force partially relieved the normal intercalation constraints. [Ru(phen)_2_dppz]^2+^ intercalated with a binding free energy of 8.2 kal/mol, which was larger than the binding free energy of the non-intercalated compounds. This can help to distinguish the intercalative mode of the ligand binding from other binding modes. The flexibility of the intercalator-saturated dsDNA was also characterized for the first time, and it was found that the persistence length of dsDNA decreased from 46 nm to 14.3 nm [29]. Finally, the binding constant (*K*_b_), binding site size (n), binding equilibrium elongation of these two ruthenium compounds, and other DNA binders obtained from various force spectroscopy experiments, are summarized in Table 2 [43,82]. Besides that, Williams et al. investigated the mechanism and binding affinity of the DNA threading intercalation kinetics with a binuclear ruthenium complex using a stretching single DNA molecule, at a range of constant stretching forces, using optical tweezers. Higher forces facilitated the intercalative binding, which led to a profound decrease in the binding site size, and resulted in one ligand intercalation at almost every DNA base stack. The force-dependent kinetics analysis revealed a mechanism that requires a DNA elongation of 0.33 nm for association, relaxation to an equilibrium elongation of 0.19 nm, and an additional elongation of 0.14 nm from the equilibrium state for dissociation [66,83]. They also reported that a ruthenium dimer complex with a flexible linker slowly threads between the DNA bases in two distinct steps under a constant applied force; the results showed that the ligand association was described by a two-step process, which consists of a fast-bimolecular intercalation of the first dppz moiety, followed by a 10-fold slower intercalation of the second dppz moiety. The second step was rate-limited by the requirement for a DNA-ligand conformational change that allows for the flexible linker to pass through the DNA duplex [84].

## 6. Conclusions

The interactions of [Ru(bpy)_2_dppz]^2+^ and [Ru(phen)_2_dppz]^2+^ with a DNA molecule have been widely studied because of their richful photophysical and photochemical properties. This review is summarized in four parts. The mechanism of the light switch benefits from the existence of two different metal-to-ligand charge transfer (MLCT) states. Both of the ruthenium complexes have an excellent selectivity to different forms and sequences of nucleotides based on the discrepancy of photoluminescence. The determination of the binding modes of the DNA-ruthenium complexes is favored for its precise understanding of the mechanism and selectivity. The binding kinetics, affinity, and magnitude of the double helix structural deformations are usually determined with the SMFS technique during the assembling of the DNA–ligand complexes at a single molecule level, which reveals new details about the molecular mechanisms governing DNA intercalation. In fact, a large number of dppz-based ruthenium complexes have been designed and synthesized in recent years, which often can be served as the modifications of [Ru(bpy)_2_dppz]^2+^ and [Ru(phen)_2_dppz]^2+^, mainly via modifications of the ancillary ligands [85,86,87,88,89,90,91,92,93,94] and dppz ligand [34,40,56,95,96,97,98]. After years of development, the dppz-based ruthenium complexes have been widely applied in many fields, such as cellular imaging [10,20,99,100,101,102,103], anticancer activity [104,105,106,107], phototherapy [108,109,110,111,112,113], protein recognition [114,115,116,117], chemosensors [32,118,119,120,121,122,123,124], and so on. However, the interaction mechanism of the modified ruthenium complexes and DNA molecule, or non-DNA-related domains at a single molecule level, is far from sufficient. Therefore, more single molecule studies are needed in order to reveal the governing assembly mechanisms that can help to guide and optimize the rational design of a new generation of antibiotic and anti-cancer drugs. The SMFS technique based on optical tweezers, atomic force microscopy, and magnetic tweezers provides an ultra-sensitive and powerful method for exploring kinetics, equilibrium constants, the energy landscape, and the binding mode of the DNA-intercalator complexes. Hence, the interaction between dppz- or related ligands-based ruthenium complexes, as well as other transition metal complexes will continue to be a fertile research area with SMFS technique.

## Figures and Tables

**Figure 1 molecules-24-00769-f001:**
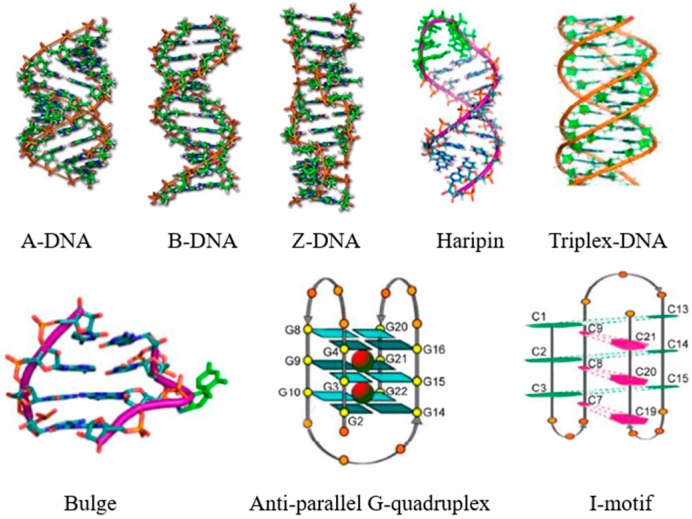
Different DNA forms. Reproduced with permission from the authors of [3,5]. Copyright © 2001, Royal Society of Chemistry. Copyright © 2014, Science China Press. B-DNA—DNA double helix. A-DNA—It is a right-handed double helix fairly similar to the more common B-DNA form, but with a shorter, more compact helical structure. Z-DNA—It is a left-handed double helical structure in which the helix winds to the left in a zigzag pattern.

**Figure 2 molecules-24-00769-f002:**
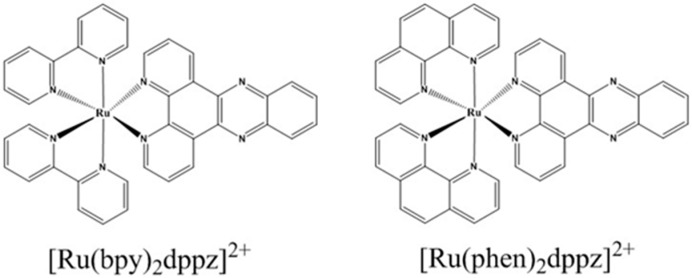
Chemical structures of intercalated ruthenium compounds. Reproduced with permission from the authors of [15]. Copyright © 1992, American Chemical Society.

**Figure 3 molecules-24-00769-f003:**
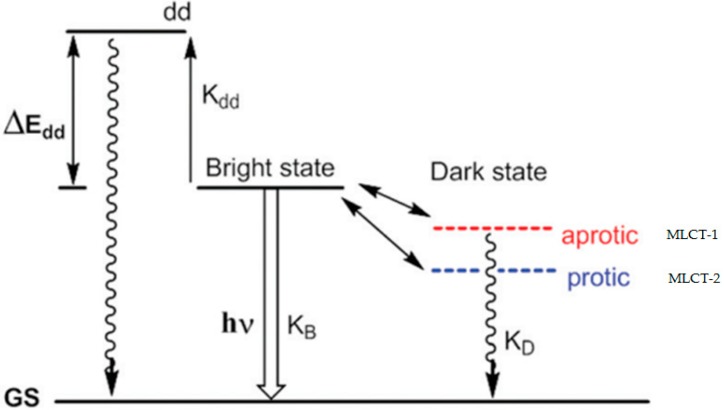
Mechanism of the molecular light switch proposed by Olson et al. Reproduced with permission from the authors of [14]. Copyright © 1997, American Chemical Society.

**Figure 4 molecules-24-00769-f004:**
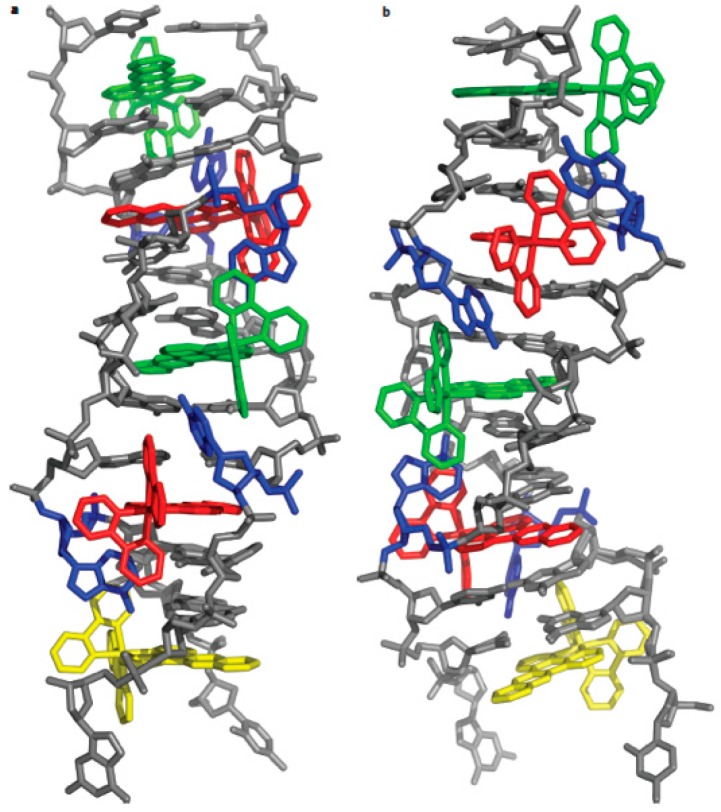
The structure of Δ-[Ru(bpy)_2_dppz]^2+^ bound to the oligonucleotide. (**a**,**b**) front view (**a**) and view rotated 90° around the helix axis (**b**). Three DNA-binding modes were observed, as follows: (1) metallo-insertion, whereby the ruthenium complex (red) inserts the dppz ligand into the DNA duplex (grey) at the mismatched sites through the minor groove, extruding the mispaired adenosines (blue); (2) metallo-intercalation, whereby the complex (green) binds between two well-matched base pairs; (3) end-capping, whereby the complex (yellow) stacked with the terminal Watson-Crick pair of the duplex. Reprinted with permission from the authors of [26]. Copyright © 2012, Springer Nature.

**Figure 5 molecules-24-00769-f005:**
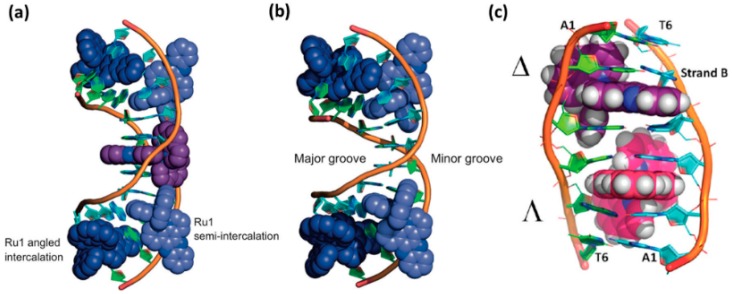
(**a**) Crystal structure of Δ-[Ru(phen)_2_dppz]^2+^ with oligonucleotide d(CCGGTACCGG)_2_. (**b**) Crystal structure of Λ-[Ru(phen)_2_dppz]^2+^ with oligonucleotide d(CCGGATCCGG)_2_. The C1C2/G9G10 intercalated Ru complex is in dark blue, and the C1C2/G9G10 semi-intercalated Ru complex is in light blue. The central AT/TA intercalated Ru complex (purple) in (**a**) is treated as a whole complex of 0.5 occupancy per oligonucleotide strand. Reprinted with permission from the authors of [27]. Copyright © 2012, Springer Nature. (**c**) Crystal structure of rac-[Ru(phen)_2_dppz]^2+^ with oligonucleotide d(ATGCAT)_2_. The Δ-[Ru(phen)_2_dppz]^2+^ is in purple, and the Λ-[Ru(phen)_2_dppz]^2+^ is in pink. Reprinted with permission from the authors of [52]. Copyright © 2013, American Chemical Society.

**Figure 6 molecules-24-00769-f006:**
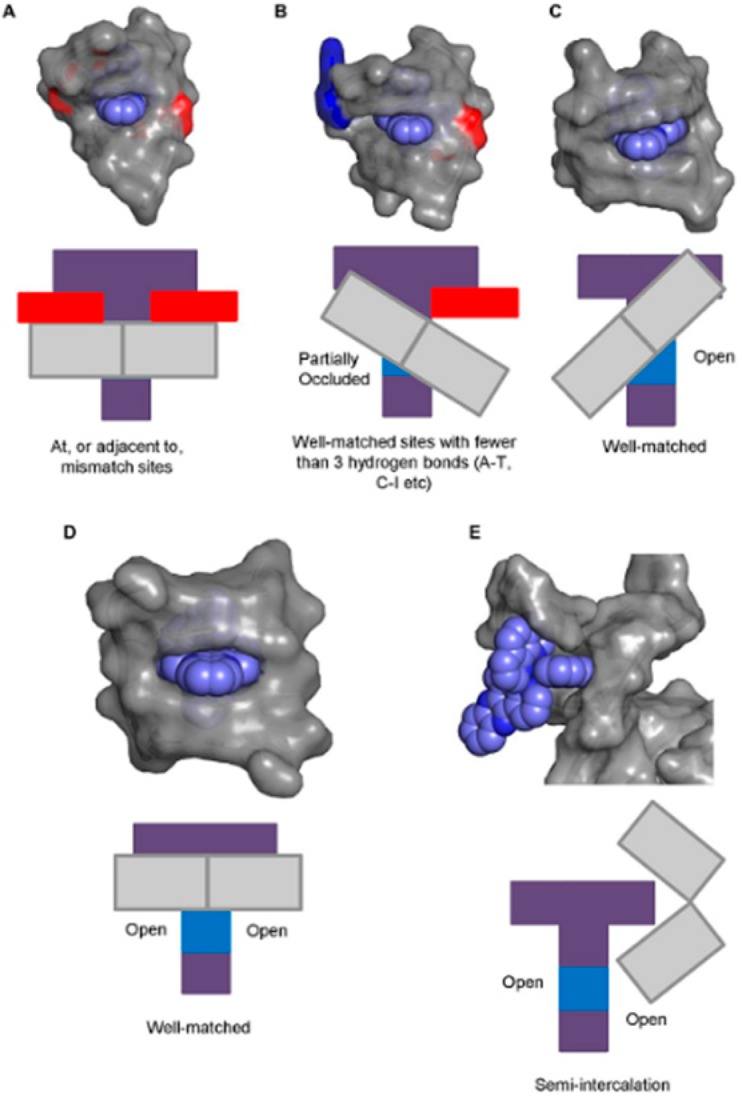
Five possible binding modes for Δ-[Ru(phen)_2_(dppz)]^2+^ to DNA. (**A**) The complex binds at, or adjacent to, a mismatch site. (**B**) Insertion into well-matched sites with less than three H-bonds between the bases. (**C**) Canted intercalation into a well-matched base pair leaves one dppz nitrogen atom entirely exposed to solvent. (**D**) Model for intercalation by a Δ enantiomer at a 5′-AT/AT-3′ step. (**E**) Semi-intercalation by an ancillary ligand into the DNA duplex, exposing both of the phenazine nitrogen atoms to the solvent. Reprinted with permission from the authors of [54]. Copyright © 2016, Oxford University Press.

**Figure 7 molecules-24-00769-f007:**
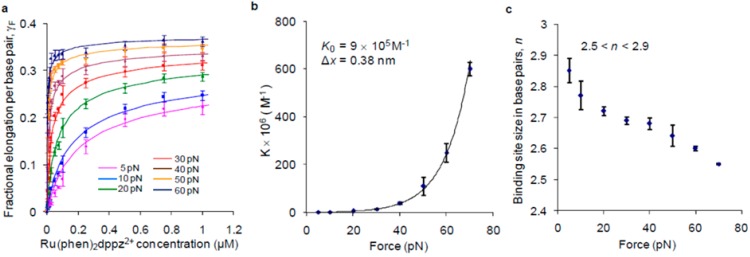
Dependence of [Ru(phen)_2_dppz]^2+^ binding on force. (**a**) Titration curve fits using the McGhee–von Hippel model. (**b**) Binding constant dependence on force. (**c**) Binding site size dependence on force. Reprinted with permission from the authors of [29]. Copyright © 2009, Springer Nature.

**Table 1 molecules-24-00769-t001:** Emission characteristics of [Ru(bpy)_2_(dppz)]^2+^ and [Ru(phen)_2_(dppz)]^2+^ upon binding to nucleic acids with different forms. Reproduced with permission from the authors of [13]. Copyright © 1992, American Chemical Society. RI: Relative emission intensity.

Nucleic Acid	[Ru(byp)_2_dppz]^2+^				[Ru(phen)_2_dppz]^2+^			
	τ (ns)	%	λ_max_ (nm)	RI	τ (ns)	%	λ_max_ (nm)	RI
poly[d(GC)]·poly[d(GC)]	220	60	610	0.29	290	60	606	0.61
	70	40			70	40		
poly(dG)·poly(dC)	260	30	610	0.29	400	40	607	0.74
	70	70			90	60		
poly[d(AT)]·poly[d(AT)]	320	20	624	0.17	740	40	620	0.75
	90	80			120	60		
poly(dA)·poly(dT)	340	40	626	0.23	840	60	621	1.39
	80	60			110	40		
poly[d(G-m^5^C)]·poly[d(d(G-m^5^C)]	240	40	606	0.25	360	40	606	0.51
	60	60			90	60		
Z-poly[d(GC)]·poly[d(GC)]	220	60	608	0.28	270	60	608	0.60
	70	40			70	40		
Calf thymus DNA Z conditions	330	40	621	0.21	750	40	616	0.80
	80	60			120	60		
poly[r(AU)]·polyp[r(AU)]	400	10	626	0.0057	490	20	620	0.10
	50	90			80	80		
poly(rG)·poly(dC)	540	10	620	0.0067	520	30	616	0.04
	70	90			80	70		
poly(dT)·poly(dA)·poly(dT)	430	70	621	0.60	530	60	621	1.45
	170	30			170	40		
tRNA	300	30	624	0.06	300	30	620	0.18
	60	70			70	70		

**Table 2 molecules-24-00769-t002:** Binding constant (K), binding site size (n), lengthening upon single intercalation (x_eq_) and twist between successive base pairs upon intercalation obtained from various force spectroscopy experiments (atomic force microscopy (or scanning force microscopy), magnetic tweezers, optical tweezers). Reproduced with permission from the authors of [43]. Copyright © 2016, Oxford University Press.

Intercalator	Binding Constant K (× 10^−6^ M^−1^)	Binding Site Size n (Base Pairs)	Binding Equilibrium Elongation ΔX_eq_ (nm/bp)
Ethidium	0.036 ± 0.005	2.01	
	10	2	
	0.46 ± 0.05	2.3 ± 0.1	0.25 ± 0.03
	0.13 ± 0.04	1.9 ± 0.1	
	0.145		
Daunomycin	0.066 ± 0.024	3.04	
AFP	2.48	2	
[Ru(phen)_3_]^2+^	0.0088 ± 0.0003	3.0 ± 0.2	
	0.0016 ± 0.0002	3.0 ± 0.1	0.28 ± 0.01
[Ru(phen)_2_dppz]^2+^	0.15 ± 0.07	2.2 ± 0.4	
	3.2 ± 0.1 (10 pN)	3.0 ± 0.5 (10 pN)	
	0.90 ± 0.10	2.9 ± 0.1	0.38 ± 0.02
[Ru(bpy)_2_dppz]^2+^	0.15 ± 0.07	2.2 ± 0.4	
	3.2 ± 0.1 (10 pN)	3.0 ± 0.5 (10 pN)	
Oxazole Yellow (YO)	0.578 ± 0.080		0.233 ± 0.013
	0.29 ± 0.09	3.8 ± 1.0	0.31 ± 0.03
Psoralen	0.088 ± 0.024	1.43 ± 0.13	
SYTOX Orange (SxO)	2.4 ± 0.5	3.0 ± 0.4	0.30 ± 0.02
SYTOX Green (SxG)	14 ± 3	2.6 ± 0.6	0.27 ± 0.02
SGold (SbG)	7.8 ± 3.3	3.2 ± 0.5	0.30 ± 0.01

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
