# Peer review of "Recent Developments in the Interactions of Classic Intercalated Ruthenium Compounds: [Ru(bpy)2dppz]2+ and [Ru(phen)2dppz]2+ with a DNA Molecule"

_molecules, 2019, doi:10.3390/molecules24040769_

Reviewer 1 Report

This is a worthwhile short summary but it definitely needs much more careful checking of the English and the references, there are all sorts of minor mistakes which i will not list here. I would advise that you get the advice of an English speaking colleague and experiences proof reader. I also recommend you look carefully at all the references you have cited since I can see that some of them have incomplete page numbers, wrong author names etc. I also hope you have obtained permission for all the diagrams you have taken from other publications and have carefully acknowledged these in the text.

This is worth publishing in spite of this, because it does represent the personal interests of the authors and a topic which can usefully be developed. 

Author Response

Dear  reviewer:

   Thank you very much for reviewing our manuscript (molecules-ID-446110) and giving the  valuable comments on the manuscript.The manuscript was revised according to the important comments, every change can be tracked with the"Track Changes" function of Microsoft Word. The new descriptions and discussions corresponding to the important comments are highlighted in revised manuscript. The point-by-point response has been attached as a Word file. We have resubmitted our revised manuscript online.

  Greatly appreciate to the important comments proposed by you that obviously improves the quality of our manuscript. Please feel free to let me know if you have any query about the changes of manuscript. I am looking forward to your response.

Best Regards

Fuchao Jia

Reviewer 2 Report

This review by Jia et al explains a detailed account of two main ruthenium complexes ranging from structure, unique characteristics, photo physics, DNA binding nature and diverse applicability in the field of molecular recognition.

The content is well structured with appropriate references, moderate changes in presentation, English language usage could be considered. When discussing the interaction of the Ruthenium complexes with DNA from the point of view of light switch effect/generation of luminescence comparison between mono-nuclear, bi-nuclear, hetero nuclear complexes could be explained from the photo physics point of view as structural changes normal affects the DNA binding characteristics of the complex. The emission characteristics upon binding to nucleic acid is nicely explained. Some more additional details with regard to minor and major groove (through the text or diagram) could be added

The differential binding capabilities of enantiomers to DNA is well explained but a comparitive account with other DNA binders could be included. Additional details with regard to binding to G quadruplex DNA could be included if possible. Single molecule force spectroscopy has been explained in a nice way. SMFS can only improve the applicability of Ruthenium metal complexes in the field of molecular recognition. Advantages over other complimentary technique with a broader view on applicability could be included.

Author Response

(The authors gave the same response as above.)
